# *Miúda* (*Nopalea cochenillifera* (L.) Salm-Dyck)—The Best Forage Cactus Genotype for Feeding Lactating Dairy Cows in Semiarid Regions

**DOI:** 10.3390/ani11061774

**Published:** 2021-06-14

**Authors:** Rubem R. Rocha Filho, Djalma C. Santos, Antonia S. C. Véras, Michelle C. B. Siqueira, Carolina C. F. Monteiro, Robert E. Mora-Luna, Lucas R. Farias, Viviany L. F. Santos, Juana C. Chagas, Marcelo A. Ferreira

**Affiliations:** 1Department of Animal Science, Federal Rural University of Pernambuco, Recife 52171900, PE, Brazil; rubem.ramos@ifal.edu.br (R.R.R.F.); antonia.veras@ufrpe.br (A.S.C.V.); michelle.siqueira2@gmail.com (M.C.B.S.); lucasthegreatr@gmail.com (L.R.F.); marcelo.aferreira@ufrpe.br (M.A.F.); 2Federal Institute of Education Science and Technology of Alagoas, Satuba 57120000, AL, Brazil; 3Agronomic Institute of Pernambuco, Experimental Station of Arcoverde, Arcoverde 56500000, PE, Brazil; djalma.cordeiro@ipa.br; 4Alagoas State University, Santana do Ipanema 57500000, AL, Brazil; monteirocarolinac@gmail.com; 5Táchira National Experimental University, San Cristóbal 5001, Táchira, Venezuela; robertmora78@yahoo.com; 6Department of Animal Science, Federal University of Piauí, Bom Jesus 64900000, PI, Brazil; vivianysantos@ufpi.edu.br; 7Department of Agricultural Research for Northern Sweden, Swedish University of Agricultural Sciences (SLU), 90183 Umeå, Sweden

**Keywords:** Cactaceae, dairy cattle, milk yield, *Nopalea*, *Opuntia*, semiarid

## Abstract

**Simple Summary:**

The usage of forage cactus is essential for the maintenance of livestock activity in semiarid regions as an alternative to conventional crops. Cactaceae have adaptive characteristics that ensure their development progress under drought conditions. Four genotypes of forage cactus (*Gigante*, *Miúda*, IPA *Sertânia*, and *Orelha de Elefante Mexicana*) were fed to lactating dairy cows and the diets were then evaluated based on animal performance, milk fatty acid profile, and microbial protein synthesis. *Miúda* forage cactus led to a higher nutrient intake and milk yield, as well as greater microbial protein synthesis. Higher saturated fatty acids were observed when the *Gigante* and IPA *Sertânia* forage cactus genotypes were fed to dairy cows. *Orelha de Elefante Mexicana* forage cactus caused lower milk yield along with protein yields and content; however, it improved the milk fatty acid profile by promoting a higher ratio of unsaturated to saturated fatty acids and desirable fatty acids. It is concluded that the *Miúda* forage cactus is the genotype most suitable for the diets of lactating dairy cows.

**Abstract:**

This study aimed to investigate the effects on nutrient intake and digestibility, milk yield (MY) and composition, milk fatty acids profile, and microbial protein synthesis caused by feeding lactating dairy cows four different forage cactus genotypes. Eight Girolando cows (5/8 Holstein × 3/8 Gyr), weighing 490 ± 69.0 kg (means ± standard deviation), and producing 15.5 ± 1.0 kg/d of milk during pretrial were distributed to two contemporaneous 4 × 4 Latin squares. The cows were fed a total mixed ration composed of sorghum silage (385 g/kg of dry matter (DM)), concentrated mix (175 g/kg DM), and forage cactus (440 g/kg DM). The experimental treatments consisted of different cactus genotypes, such as *Gigante* cactus (GC), *Miúda* cactus (MC), IPA *Sertânia* cactus (SC), and *Orelha de Elefante Mexicana* cactus (OEMC). The feeding of MC provided a higher intake of DM, organic matter (OM), and total digestible nutrients, as well as higher MY, energy-corrected milk, and microbial protein synthesis in comparison with those resulting from the other genotypes tested. The GC promoted lower DM and OM, and the apparent digestibility of neutral detergent fiber. The cows fed with OEMC showed lower MY and milk protein yield and content, and higher unsaturated over saturated fatty acid ratio in milk. *Miúda* forage cactus increased nutrient intake, digestibility of DM and OM, and microbial synthesis without impairing the milk fatty acid profile.

## 1. Introduction

Despite the adverse climate in semiarid regions [1], dairy farming is one of the most important economic alternatives for smallholder farmers who use family labor. The usage of forage cactus has proved essential for the maintenance of livestock activity in these regions, as it has adaptive characteristics which ensure that its development progresses under arid conditions [2]. In fact, the forage cactus is deemed to be the queen of forage crops in dryland areas due to its high nutritional value and the energy and water content produced per unit area, compared with conventional crops recommended for semiarid regions (e.g., sorghum silage, buffel grass, and corn silage) [3]. Additionally, due to its high moisture content, forage cactus also meets most of the nutritional needs of the animals, minimizing a major livestock problem in these regions [4].

In Brazil, the most used genotypes of forage cactus are *Gigante* cactus (GC), *Redonda* cactus (*Opuntia*
*fícus-indica* Mill), and *Miúda* cactus (MC; *Nopalea cochenillifera* (L.) Salm-Dick), with an estimated planted area of 500,000 ha. The Agronomic Institute of Pernambuco (IPA) and the Federal Rural University of Pernambuco have selected clones of forage cactus, such as IPA-100004 (*Miúda)*, IPA-200205 (*IPA Sertânia cactus* (SC); *N. cochenillifera* (L.) Salm-Dick), and IPA-200016 (*Orelha de Elefante Mexicana cactus* (OEMC); *Opuntia stricta* (Haw.)), which stand out in terms of agronomic performance [5,6,7]. However, except for MC and GC, little is known about animal performance when other genotypes are used. It has been noted that evaluation of forages cannot be performed without the use of animals; thus, the results are representative of the existing conditions of production systems [8].

The present study aimed to evaluate the effects on nutrient intake and digestibility, milk yield (MY) and composition, milk fatty acid (FA) profile, and microbial protein synthesis caused by feeding lactating dairy cows in semiarid regions four different genotypes of forage cactus (GC, MC, SC, and OEMC).

## 2. Materials and Methods

### 2.1. Animals and Diets

The experiment was conducted at the Agronomic Institute of Pernambuco (IPA), the Arcoverde Experimental Station, Arcoverde, Brazil, in accordance with the guidelines and recommendations of the Committee of Ethics on Animal Studies at the Federal Rural University of Pernambuco (License N° 069/2016). Eight Girolando cows (5/8 Holstein × 3/8 Gyr) with an average body weight of 490 kg ± 69.0 kg (mean ± standard deviation), milk production of 15.5 ± 0.4 kg/day, and eight weeks of lactation pretrial were distributed to two contemporaneous 4 × 4 Latin squares and assigned to four experimental treatments. Each experimental period lasted 21 days, with the first 14 days allowing for adaptation to the diet and the remaining 7 days used for evaluation and sample collection. The cows were kept in individual pens equipped with a feed bunk and freshwater source.

The chemical composition and nutritional value of the dietary ingredients are shown in Table 1. The experimental treatments consisted of a total mixed ration containing a genotype of forage cactus, sorghum silage, soybean meal, urea, and a mineral mixture (Table 2). The diets were formulated according to the recommendations from the NRC [9] to meet the requirements of lactating cows with a body weight of 500 kg and a MY of 14 kg/day with 4% fat.

The total mixed ration was provided ad libitum twice daily (60% at 07:00 h and 40% at 15:30 h), allowing leftovers of 5–10% of total dry matter (DM) offered. All forage cactus used was purchased from the IPA experimental station. The sorghum silage was made at the experimental station itself. The forage cacti were chopped in a forage machine immediately before being supplied to the animals and were mixed with other ingredients in the trough.

### 2.2. Nutrient Intake and Digestibility Assay

The diet offered and leftovers were weighed daily, from the 15th to the 21st day of the experimental period, to estimate the nutrient intake. Samples of feeds and leftovers were pre-dried in a forced-air oven at 60 °C until they reached a constant weight, and a sample was made per animal for subsequent chemical analysis.

To calculate the apparent nutrient digestibility, the fecal DM production of the cows was estimated using the external marker chromic oxide. A total of 20.0 g of the marker was placed in a wrapper and provided orally, twice a day, at the morning and evening feeding times for 12 days. Seven of these days were used for the adaptation and regulation of the marker flow, and five were used to perform sampling. The fecal samples were collected directly from the rectum twice daily after milking in the morning and afternoon. The samples from each cow in each period were dried in a forced-air oven at 60 °C for 48 h and ground in a Wiley mill with a 1 mm sieve for the subsequent formation of composite samples.

### 2.3. Milk Yield and Chemical Composition

The animals were manually milked twice daily (05:30 and 14:00 h), and the milk yield (MY) was individually recorded. Milk samples were collected from each animal on the 6th and 7th days of collection in an amount proportional to the morning and afternoon productions to form the composite sample. Part of the composite sample (50 mL) was stored in a Bronopol container and sent to the PROGENE laboratory (Dairy Cattle Management Program of the Northeast) to determine the level of fat, protein, lactose, and total dry extract. Another fraction of the composite sample was deproteinized with 25% trichloroacetic acid (10 mL milk:5 mL acid), filtered through filter paper, and stored at −20 °C for subsequent analysis of allantoin and milk urea nitrogen (MUN) conducted in the filtrate.

### 2.4. Plasma Urea Nitrogen and Urine Collection

Blood samples were collected on the 12th day of each experimental period, four hours after the morning feeding, using heparin as an anticoagulant. These samples were subsequently centrifuged at 3130× *g* for 10 min. The resulting plasma was stored at −20 °C for subsequent plasma urea nitrogen (PUN) analysis. On this occasion, a ‘spot’ urine sample from each cow was collected, homogenized, and filtered; a 10 mL aliquot was also extracted. These aliquots were diluted in 40 mL of 0.036 N sulfuric acid and stored at −20 °C for subsequent analysis of creatinine, urea nitrogen, uric acid, and allantoin.

### 2.5. Chemical Analyses

Samples of feed, leftovers, and feces were analyzed for DM (method 934.01), organic matter (OM; method 942.05), crude protein (CP; nitrogen × 6.25; method 984.13), and ether extract (EE; method 920.39) according to the AOAC [10] procedures. Analyses of neutral detergent fiber (NDF) were performed according to Mertens [11] using thermostable α-amylase without the use of sodium sulfite. Analyses of acid detergent fiber and lignin (method 973.18) were performed according to AOAC [10]. Total sugars were measured using the Lane–Eynon method, and the starch was measured by the acid hydrolysis method, as described by AOAC [10]. Measurements of neutral detergent insoluble protein, acid detergent insoluble protein, and nonprotein nitrogen were performed according to Licitra et al. [12]. Estimates of indigestible DM and indigestible NDF of the forage cactus genotypes were performed using in situ incubation for 288 h, as proposed by Valente et al. [13]. To determine the chromic oxide content, the fecal samples were analyzed by atomic absorption spectrophotometry according to Fenton and Fenton [14].

For the extraction of milk FA, the technique described by Murphy et al. [15] was used, which incorporates freezing the milk, thawing it, and then performing centrifugation. Methyl esters of FA were obtained by the transesterification of triacylglycerols, according to the methodology described by the International Organization of Standardization [16], using n-heptane and KOH/methanol. Esters of FA were quantified using a gas chromatograph (model CG-Master, manufactured by Ciola and Gregori, São Paulo, Brazil) equipped with a flame ionization detector and a fused silica capillary column with a length of 100 m and an internal diameter of 0.25 mm, along with 0.25 µm cyanopropyl polysiloxane. A carrier gas (H_2_) with a flow of 1.2 mL/min, 30 mL/min of N_2_, and 30 and 300 mL/min of H_2_ and synthetic air, respectively, were used for the flame of the detector. An injected volume of 1 µL was used with a 1:100 split. Temperatures of 220 °C for the injector and 230 °C for the detector were used. The initial column temperature of 60 °C was maintained for seven minutes, then raised to 140 °C at a rate of 40 °C/min and maintained for 20 min; finally, it was raised to 225 °C at a rate of 5 °C/min and maintained for 15 min. The identification of FA was performed by comparing the retention times obtained by Sigma-Aldrich (St. Louis, MO, USA) standards with the concentrations obtained by the calculation of the peak areas.

Measurements of urea nitrogen, creatinine, and uric acid were performed using commercial kits (Doles) according to the manufacturer’s recommendations. Allantoin determinations were made using the colorimetric method, as described by Chen and Gomes [17], and the absorbance was measured on a Bel Engineering^®^ spectrophotometer (Model SP 2000UV, Monza, Italy).

### 2.6. Calculations

The total carbohydrates (TCH) were estimated according to Sniffen et al. [18], while nonfibrous carbohydrate (NFC) contents were estimated according to Detmann and Valadares Filho [19]. Organic matter was calculated (OM = 1000 g/kg DM-g ash/kg DM). Dry matter, OM, CP, NDF, TCH, and NFC intakes were calculated by subtracting their amount contained in the refusals from the daily amounts offered.

The DM fecal production was estimated using the following equation: FP = amount of marker provided (g/day)/marker in fecal DM (g/kg). The apparent nutrient digestibility was estimated using the following equation: (nutrient intake − fecal nutrient excreted)/nutrient intake. The total digestible nutrients (TDN) were estimated according to Sniffen et al. [18].

Energy-corrected milk (ECM) was calculated according to Orth [20]: (0.327 × milk yield (kg/d)) + (12.95 × fat yield (kg/d)) + (7.2 × protein yield (kg/d)). Feed efficiency was calculated from the relationship between the ECM and the DM intake for each cow in each experimental period.

The daily urinary excretion was estimated from a proposed excretion of creatinine of 24.05 mg/kg of body weight [21]. Microbial protein synthesis was estimated by determining the urinary and milk excretion of purine derivatives (allantoin and uric acid). To determine the amount of microbial purine absorbed (X mmol/day) from the excretion of purine derivatives (Y mmol/day), the following equation was used: Y = 0.85 × X + (0.385 × body weight^0.75^) [22]. The intestinal flow of microbial nitrogen was calculated from the microbial purine absorbed, according to Chen et al. [23].

### 2.7. Statistical Analysis

The studied variables were subjected to ANOVA using the MIXED procedure of SAS (Version 9.4; SAS Inst., Inc., Cary, NC, USA) according to a replicated 4 × 4 Latin square design. The following mathematical model was used:Y_ijkl_ = µ + A_i_ (Q)_k_ + P_j_ + Q_k_ + G_l_ + (G × Q)_lk_ + e_ijkl_(1)
where Y_ijkl_ is the observed variable; µ is the population mean; A_i_ (Q)_k_ is the random effect of animals within squares; P_j_ is the random effect of the period; Q_k_ is the fixed effect of the square; G_l_ is the fixed effect of forage cactus genotype; (G × Q)_lk_ is the interaction between forage cactus genotype and square; and e_ijkl_ is the random residual error.

Differences among forage cactus genotypes were declared statistically significant at *p* < 0.05, and Tukey’s test was used.

## 3. Results

### 3.1. Chemical Composition of Feed and Diets

The evaluated genotypes of the forage cactus had low levels of DM, indigestible DM, CP, and NDF, but high levels of NFC (Table 1). The GC showed higher OM and NFC contents than the other genotypes, while the MC showed higher starch and sugar contents. The CP content of the OEMC was higher than that of the other genotypes, and the diets had similar chemical compositions (Table 2).

### 3.2. Nutrient Intake and Apparent Digestibility

The diets composed of MC promoted higher (*p* < 0.01) intakes of DM, OM, CP, NDF, and TCH (Table 3). The SC promoted higher (*p* < 0.01) CP intake but lower (*p* < 0.01) intakes of EE and NFC. When the OEMC was supplied, lower (*p* < 0.01) intakes of OM, EE, TCH, and NFC were observed.

The MC showed higher (*p* < 0.01) digestibility of DM, OM, and TCH than the other genotypes (Table 4). The digestibility of CP was lower for the GC and OEMC. The GC showed the lowest (*p* < 0.01) NDF digestibility as compared to the other genotypes. No differences (*p* ≥ 0.24) were observed for EE and NFC digestibility.

### 3.3. Milk Yield and Chemical Composition

Dairy cows that were fed diets with MC and GC showed higher (*p* ≤ 0.04) MY, ECM, and milk protein yield, and cows fed with only MC showed a higher protein milk concentration (Table 5). Lower values for those variables were observed when OEMC was tested. The yields of protein, lactose, and total solids were lower (*p* ≤ 0.01) when dairy cows were fed SC and OEMC. Furthermore, the lowest (*p* ≤ 0.01) feed efficiency was observed when the cows were fed only MC.

The proportions of the saturated fatty acids (SFA) C_4:0_, C_10:0_, and C_12:0_ were lower (*p* < 0.01) for the SC (Table 6). However, it showed a higher (*p* < 0.01) proportion of C_16:0_, whereas a greater (*p* < 0.01) proportion of C_15:0_ was observed for the GC. Greater proportions of C_16:1_, C_18:0_, and C_18:1_ were observed for the OEMC. Moreover, the OEMC promoted a lower (*p* = 0.02) SFA proportion, a higher (*p* ≤ 0.03) unsaturated fatty acids (UFA) proportion and UFA:SFA ratio, as well as a higher desirable fatty acid (DFA) proportion.

### 3.4. Urea Nitrogen and Microbial Protein Synthesis

Dairy cows fed with SC and OEMC showed higher (*p* < 0.01) levels of PUN, but lower (*p* < 0.015) concentrations were observed when GC was used, as well as a lower excretion of urinary urea nitrogen (UUN) (Table 7). The SC and OEMC promoted a higher (*p* < 0.01) concentration of MUN compared with GC and MC. Greater purine derivatives excretion and microbial protein synthesis were observed for MC and lower for OEMC (*p* = 0.04). No differences among forage cactus genotypes were observed (*p* = 0.96) in the microbial protein synthesis efficiency.

## 4. Discussion

The compositions observed for the cactus genotypes are consistent with those presented in other studies performed with the genera *Nopalea* and *Opuntia* [24,25,26,27]. The indigestible DM and NDF values, such as those from elephant grass, sugarcane, and corn silage, are lower than those from forages [13]. Diets showed similar metabolizable energy contents because of the similar metabolizable energy content of the forage cactus genotypes, with values of 2.15 Mcal/kg DM (GC), 2.17 Mcal/kg DM (MC), 2.13 Mcal/kg DM (SC), and 2.15 Mcal/kg DM (OEMC) [3].

During the present experiment, animals receiving MC demonstrated higher appetites. The highest levels of sugars and starch observed for this forage cactus may contribute to the high palatability attributed to this genotype [28], partially justifying the higher DM intake provided. Inácio et al. [25] also observed greater DM intake with MC than OEMC.

The inclusion of higher levels (320–500 g/kg DM) of forage cactus in diets for lactating dairy cows [29] and dairy heifers [30] has been demonstrated to reduce NDF digestibility, which has been associated with the increase in NFC content in the diet, resulting in a possible decrease in ruminal pH and/or increase in the rate of passage. Although the levels of inclusion of the forage cactus genotypes were the same, the GC promoted higher NFCI:DMI and NFCI:NDFI ratios, which could explain the lower NDF digestibility. Zebeli et al. [31] reported that the ruminal pH and rate of passage explained 62.0% of the variation in NDF digestibility of the diet of lactating cows. Zebeli et al. [32] analyzed research data and observed a negative linear effect between the NFC:NDF ratio and ruminal pH. They also noted that the NDF digestibility linearly decreased with the increase of the NFC:NDF ratio in the diet of dairy cows.

Forage cactus is rich in carbohydrates suited for rapid fermentation, with over 80% of DM disappearing within 48 h of ruminal incubation, and the GC have presented a greater gas potential production in vitro than the MC genotype [33]. High dietary levels of rapid fermentation carbohydrates can result in lower ruminal pH, reducing fiber degradation due to the decreased population or activity of the cellulolytic microorganisms [34], for which pH values between 6.00 and 6.10 are considered a threshold [35]. The lower NDF digestibility observed when GC was provided is in accordance with the findings of Rocha Filho et al. [3], who observed lower NDF digestibility for GC when five genotypes of cactus were evaluated: GC, MC, SC, *Orelha de Elefante Africana* (*Opuntia*), and OEMC in sheep diets at 400 g/kg DM. Moreover, they observed lower ruminal pH when GC was provided, with pH values between 6.10 and 6.26 after feeding, despite NFC intake being similar among the cactus genotypes.

Conversely, changes in ruminal pH and/or fermented substrate, which cause changes in the microbial population, may have negative consequences for the ruminal protein degradation, which depends on the action of proteolytic and nonproteolytic enzymes. Reduced ruminal fiber digestion can influence protein degradation through the reduction of microbial access to protein, which is linked to the fiber fraction [36]. This could partially explain the lower digestibility of protein in diets with GC. Furthermore, under conditions of lower ruminal pH, the amount of protein absorbed in the small intestine is reduced [34]. As there is no absorption of this protein, there may be increased excretion of fecal nitrogen, with a consequent decrease in digestibility.

One must consider that NFC has different components, such as nonstructural carbohydrates (sugars and starches), neutral detergent soluble fiber (fructans, β-glucans, and pectin), and organic acids, and that they may generate different patterns of ruminal fermentation [37]. The profile of the NFC diet could affect the supply of metabolizable nutrients for the animal and change the MY and composition with variable results [38]. Thus, the lower MY and ECM observed when OEMC was supplied could be the result of the ruminal fermentation pattern, resulting in lower availability of metabolizable nutrients, given that there was no difference in the TDN intake among the experimental diets. Furthermore, the GC promoted a MY and ECM similar to that of the MC, despite the lower TDN intake. Milk yield results of the current study should be interpreted with caution, as the number of cows used in a nontriplicated 4 × 4 Latin square might have compromised the statistical power of the results obtained.

The lower OM intake might have also negatively influenced the MY observed for OEMC due to the lower availability of fermentable material in the rumen. The feeding efficiency of GC and OEMC was greater than that of MC due to the lower DM intake promoted by these diets. The fat contents observed, above 40 g/kg for all treatments, indicate that an adequate level of effective fiber was maintained in the diet, and that normal rumen function was maintained [39] despite possible differences in the fermentation pattern.

The higher milk protein content observed when MC was provided could be related to higher microbial protein synthesis in the rumen. Microbial protein contributes to a large amount of CP, which passes to the small intestine where greater amino acid flow could increase the milk components [40]. The lower daily production of protein, lactose, and total solids observed when OEMC was provided mainly resulted from the lower milk production promoted by this diet.

Concerning milk FA, the OEMC promoted desirable changes. In order to reduce possible harm to human health, a decrease in the intake of SFA from foods of animal origin is recommended, especially lauric (C_12:0_), myristic (C_14:0_), and palmitic (C_16:0_) acids, which have been linked to cardiovascular problems. Stearic acid (C_18:0_), although saturated, has no effect on blood cholesterol levels [41]. Thus, OEMC promoted a higher UFA:SFA ratio and a higher proportion of DFA in the milk, which could be considered beneficial for the health of people consuming this product.

In the case of the SC, there was a higher C_16:0_ content and a lower C_18:0_ content in the milk, which can be considered unfavorable, reducing the UFA:SFA ratio and DFA proportion in the milk fat. Because half of the milk FA with 12–16 C originates from de novo synthesis in the mammary gland, and the other half (as well as all the FA with over 18 C) originates mainly from the diet, their concentration in the milk can be influenced by the FA present in the feed provided. However, it should be considered that the extensive biohydrogenation of 18 C UFA is performed by rumen microorganisms, decisively influencing the proportion of C_18:0_ in milk fat [42].

The lowest PUN concentrations observed when GC was provided could be partly explained by the lower intake of dietary CP [43]. In the case of SC and OEMC, which provided more CP than GC, the higher PUN concentrations could indicate a lower efficiency in the utilization of the consumed CP [44]. The lower MUN produced by GC and MC suggests a lower ruminal degradable protein supply. However, cacti and diets showed a similar CP concentration (Table 1), and its effective degradability is also similar among cacti (640–690 g/kg DM) [45]. In addition, DM effective degradability for GC, MC, and SC does not differ (700–712 g/kg DM) [46]. In contrast, the intermediately degradable carbohydrate fraction varies among species and genotypes of cacti, while rapidly and slowly degradable carbohydrate fractions do not differ [33]. Therefore, the intermediately degradable carbohydrate fraction may be responsible for the differences of MUN between cacti.

One must consider that the efficiency of the usage of nitrogen by ruminal microorganisms may be influenced by the amount, type, and degradability of carbohydrates and protein present in the diet [47]. Changes in the degradation and ruminal fermentation of energy and protein sources can cause a loss of nitrogen as ammonia, or a decrease in the microbial protein synthesis [48]. The increased excretion of UUN when MC, SC, and OEMC were used could be an indication of dietary CP intake above the requirements for the production level presented by the animals [49]. Higher UUN excretion is related to a higher concentration of PUN as there is a positive relationship between these variables [43].

The higher purine derivative excretion and microbial protein synthesis promoted by MC might be a consequence of higher OM, CP, NDF, and NFC intake, which led to a better energy-to-protein ratio in the rumen. The CP, NFC, and FDN are dietary components which can potentially be manipulated to optimize ruminal fermentation and increase the passage of amino acids to the small intestine [40]. The NRC [9] recommends a microbial protein synthesis efficiency of 130 g microbial CP per kg of consumed TDN. Valadares Filho et al. [50] proposed adopting 120 g microbial CP per kg of consumed TDN as a reference value for tropical conditions, which is close to the values observed in the present study. Thus, the results indicate that the use of forage cactus genotypes did not limit the synthesis and efficiency of microbial protein synthesis.

Among the forage cactus evaluated in this study, MC was superior due to its nutritional value, promoting improvement in animal performance as compared to the other genotypes. However, feeding SC and OEMC to dairy cows also supported satisfactory animal performance. Therefore, the SC and OEMC genotypes may also be recommended for feeding lactating dairy cows reared in semiarid regions, in situations where other factors are considered (e.g., agronomic performance).

## 5. Conclusions

*Miúda* forage cactus increased intake of nutrients, digestibility of DM and OM, and microbial synthesis without impairing the FA profile of milk. Therefore, MC is recommended as the most suitable forage cactus genotype for feeding lactating dairy cows reared in semiarid regions. Furthermore, the supply of SC and OEMC also promoted satisfactory animal performance; thus, these genotypes may be adopted as alternatives to MC.

## Figures and Tables

**Table 1 animals-11-01774-t001:** Chemical composition of the experimental dietary ingredients (g/kg of DM, unless otherwise stated).

Item ^1^	Forage Cactus				Sorghum Silage	Soybean Meal
	Gigante	Miúda	IPA Sertânia	Orelha de Elefante Mexicana		
Dry matter	87	96	77	75	301	879
OM	929	846	831	869	934	929
CP	46	50	57	66	54	486
NPN	5.3	7.0	3.5	8.7	ND	ND
EE	23	17	15	20	23	22
NDF	252	244	241	272	679	178
ADF	133	139	139	135	397	102
Lignin	34	16	15	29	64	5.4
TCH	860	779	760	783	858	421
NFC	608	536	519	511	179	245
Total sugars	93	137	90	115	ND	ND
Starch	143	252	165	135	ND	ND
NDIP	6.3	7.4	6.0	7.4	15	27
ADIP	2.5	2.2	2.1	2.0	6.4	2.0
iDM	92	96	85	97	ND	ND
iNDF	72	63	60	80	ND	ND

^1^ OM—Organic matter; CP—Crude protein; NPN—Non-protein nitrogen; EE—Ether extract; NDF—Neutral detergent fiber; ADF—Acid detergent fiber; TCH—Total carbohydrates; NFC—Nonfibrous carbohydrates; NDIP—Neutral detergent insoluble protein; ADIP—Acid detergent insoluble protein; iDM—Indigestible dry matter; iNDF—Indigestible neutral detergent fiber; ND—not determined.

**Table 2 animals-11-01774-t002:** Ingredients and chemical composition of the experimental diets (g/kg of DM unless otherwise stated).

Ingredients	Forage Cactus ^3^			
	GC	MC	SC	OEMC
*Gigante*	440	-	-	-
*Miúda*	-	440	-	-
*IPA Sertânia*	-	-	440	-
*Orelha de Elefante Mexicana*	-	-	-	440
Sorghum silage	385	385	385	385
Soybean meal	150	150	150	150
Urea	8	8	8	8
Mineral and vitamin supplement	17	17	17	17
Chemical composition				
Dry matter	154	166	139	136
Organic matter	916	879	873	889
Crude protein	137	138	141	145
Ether extract	22	19	18	21
Neutral detergent fiber	399	395	394	408
Acid detergent fiber	226	229	229	227
Total carbohydrates	757	721	713	723
Nonfibrous carbohydrates	358	326	319	316
Total digestible nutrients ^1^	582	629	614	593
Metabolizable energy (Mcal/kg DM) ^2^	2.10	2.27	2.22	2.14

^1^ Obtained from digestibility assay; ^2^ Calculated as TND × 4.409 × 0.82 [9]; ^3^ GC—*Gigante* cactus; MC—*Miúda* cactus; SC—IPA *Sertânia* cactus; OEMC—*Orelha de Elefante Mexicana* cactus.

**Table 3 animals-11-01774-t003:** Effects of forage cactus genotypes on nutrient intake (*n* = 8).

Item (kg/day) ^1^	Forage Cactus ^2^				SEM ^3^	*p*-Value
	GC	MC	SC	OEMC		
Dry matter	13.1 ^b^	14.9 ^a^	13.6 ^a,b^	12.1 ^b^	0.73	<0.01
Organic matter	11.9 ^a,b^	13.1 ^a^	11.9 ^a,b^	10.8 ^b^	0.66	<0.01
Crude protein	1.86 ^b^	2.18 ^a^	2.10 ^a^	1.97 ^a,b^	0.027	<0.01
Neutral detergent fiber	4.96 ^b^	5.58 ^a^	5.28 ^a,b^	4.92 ^b^	0.262	<0.01
Ether extract	0.29 ^a^	0.29 ^a^	0.25 ^b^	0.26 ^b^	0.015	<0.01
Total carbohydrates	9.80 ^a,b^	10.6 ^a^	9.57 ^b^	8.58 ^c^	0.544	<0.01
Nonfibrous carbohydrates	4.84 ^a,b^	5.08 ^a^	4.29 ^b^	3.67 ^c^	0.329	<0.01
Total digestible nutrients	7.92 ^b^	9.51 ^a^	8.44 ^a,b^	7.25 ^b^	0.610	<0.01
NFC/DM (kg/kg)	0.37 ^a^	0.34 ^b^	0.31 ^c^	0.30 ^c^	0.011	<0.01
NFC/NDF (kg/kg)	0.98 ^a^	0.91 ^a^	0.81 ^b^	0.74 ^b^	0.044	<0.01

^1^ DM—dry matter; NDF—Neutral detergent fiber; NFC—Nonfibrous carbohydrates. ^2^ GC—*Gigante* cactus; MC—*Miúda* cactus; SC—IPA *Sertânia* cactus; OEMC—*Orelha de Elefante Mexicana* cactus. ^3^ SEM—Standard error of the mean. ^a,b,c^ Means with different superscripts in the same row are significantly different (*p* < 0.05).

**Table 4 animals-11-01774-t004:** Effects of forage cactus genotypes on apparent digestibility of dietary chemical components (g/kg; *n* = 8).

Item	Forage Cactus ^1^				SEM ^2^	*p*-Value
	GC	MC	SC	OEMC		
Dry matter	582 ^c^	667 ^a^	652 ^a,b^	608 ^b,c^	18.1	<0.01
Organic matter	611 ^c^	693 ^a^	679 ^a,b^	637 ^b,c^	16.3	<0.01
Crude protein	748 ^b^	819 ^a^	823 ^a^	782 ^b^	12.2	<0.01
Ether extract	554	593	592	606	30.0	0.56
Neutral detergent fiber	342 ^b^	489 ^a^	494 ^a^	431 ^a^	28.8	<0.01
Total carbohydrates	596 ^c^	680 ^a^	659 ^a,b^	615 ^b^	17.5	<0.01
Non fibrous carbohydrates	856	892	865	868	13.5	0.24

^1^ GC—*Gigante* cactus; MC—*Miúda* cactus; SC—IPA *Sertânia* cactus; OEMC—*Orelha de Elefante Mexicana* cactus. ^2^ SEM: standard error of the mean. ^a,b,c^ Means with different superscripts in the same row are significantly different (*p* < 0.05).

**Table 5 animals-11-01774-t005:** Effects of forage cactus genotypes on milk production and composition, and feed efficiency (*n* = 8).

Item ^1^	Forage Cactus ^2^				SEM ^3^	*p*-Value
	GC	MC	SC	OEMC		
Yield (kg/d)						
Milk	13.6 ^a^	13.5 ^a^	12.9 ^ab^	12.7 ^b^	0.547	<0.01
ECM	15.0 ^a^	15.2 ^a^	14.3 ^a,b^	13.9 ^b^	0.477	0.01
Fat	0.570	0.580	0.550	0.53	0.025	0.07
Protein	0.450 ^a^	0.450 ^a^	0.410 ^b^	0.400 ^b^	0.012	<0.01
Lactose	0.630 ^a^	0.620 ^a,b^	0.590 ^b,c^	0.580 ^c^	0.031	<0.01
Total solids	1.79 ^a^	1.79 ^a^	1.69 ^a,b^	1.64 ^b^	0.059	<0.01
Composition (g/kg)						
Fat	42.4	43.0	43.0	42.0	0.241	0.77
Protein	33.1 ^a,b^	33.3 ^a^	32.1 ^a,b^	31.8 ^b^	0.019	0.04
Lactose	46.0	45.8	45.5	45.8	0.051	0.39
Total solids	132	133	131	130	0.329	0.22
Feed efficiency	1.17 ^a^	1.02 ^b^	1.06 ^a,b^	1.15 ^a^	0.046	<0.01

^1^ ECM—energy-corrected milk. ^2^ GC—*Gigante* cactus; MC—*Miúda* cactus; SC—IPA *Sertânia* cactus; OEMC—*Orelha de Elefante Mexicana* cactus. ^3^ SEM: standard error of the mean. ^a,b,c^ Means with different superscripts in the same row are significantly different (*p* < 0.05).

**Table 6 animals-11-01774-t006:** Effects of forage cactus genotypes on milk fatty acid profile (g/kg of fatty acids; *n* = 8).

Item ^1^	Forage Cactus ^2^				SEM ^3^	*p*-Value
	GC	MC	SC	OEMC		
4:0	33.7 ^a^	28.6 ^a^	16.1 ^b^	27.5 ^a^	2.79	<0.01
6:0	10.1	6.38	8.39	8.61	1.43	0.33
8:0	7.78	5.03	5.86	5.60	1.02	0.28
10:0	42.4 ^a^	33.9 ^a,b^	16.3 ^b^	38.3 ^a^	4.86	<0.01
12:0	66.7 ^a^	60.3 ^a^	25.0 ^b^	57.7 ^a^	5.48	<0.01
14:0	173 ^a,b^	194 ^a^	197 ^a^	158 ^b^	9.01	0.01
14:1	9.40	5.79	8.91	7.73	1.47	0.33
15:0	15.7 ^a^	7.20 ^b^	8.91 ^b^	9.18 ^b^	1.40	<0.01
16:0	384 ^b^	395 ^b^	492 ^a^	335 ^b^	17.6	<0.01
16:1	17.1 ^a,b^	8.45 ^b^	12.8 ^a,b^	23.9 ^a^	3.41	<0.01
17:0	6.16	3.40	5.84	6.13	0.840	0.08
18:0	58.3 ^a,b^	36.6 ^b^	28.7 ^b^	91.2 ^a^	8.91	<0.01
18:1	168 ^b^	208 ^a,b^	165 ^b^	221 ^a^	13.5	0.01
18:2	6.64	5.15	8.09	8.86	1.29	0.21
SFA	798 ^a^	773 ^a,b^	805 ^a^	738 ^b^	14.8	0.02
UFA	202 ^b^	227 ^a,b^	195 ^b^	261 ^a^	14.8	0.02
UFA:SFA	0.26 ^b^	0.30 ^a,b^	0.25 ^b^	0.36 ^a^	0.020	0.01
DFA	260 ^a,b^	264 ^a,b^	224 ^b^	353 ^a^	21.1	0.03

^1^ SFA—Saturated fatty acids; UFA—Unsaturated fatty acids; DFA—Desirable fatty acids (UFA + 18:0). ^2^ GC—*Gigante* cactus; MC—*Miúda* cactus; SC—IPA *Sertânia* cactus; OEMC—*Orelha de Elefante Mexicana* cactus. ^3^ SEM: standard error of the mean. ^a,b^ Means with different superscripts in the same row are significantly different (*p* < 0.05).

**Table 7 animals-11-01774-t007:** Effects of forage cactus genotypes on urea excretion and microbial protein synthesis (*n* = 8).

Item ^1^	Forage Cactus ^2^				SEM ^3^	*p*-Value
	GC	MC	SC	OEMC		
PUN (mg/dL)	6.28 ^c^	8.49 ^b^	12.4 ^a^	13.0 ^a^	0.597	<0.01
MUN (mg/dL)	6.04 ^b^	7.01 ^b^	10.6 ^a^	9.70 ^a^	0.711	<0.01
UUN (g/day)	74.2 ^b^	137 ^a^	157 ^a^	171 ^a^	12.8	<0.01
PDE (mmol/day)	203 ^a,b^	233 ^a^	208 ^a,b^	191 ^b^	10.5	0.04
MPS (g/day)	857 ^a,b^	1029 ^a^	898 ^a,b^	805 ^b^	57.5	0.04
MPSE (g/kg TDN)	108	111	109	112	6.46	0.96

^1^ PUN—Plasma urea nitrogen; MUN—milk urea nitrogen; UUN—Urine urea nitrogen; PDE—Purine derivatives excretion; MPS—Microbial protein synthesis; MPSE—Microbial protein synthesis efficiency. ^2^ GC—*Gigante* cactus; MC—*Miúda* cactus; SC—IPA *Sertânia* cactus; OEMC—*Orelha de Elefante Mexicana* cactus. ^3^ SEM: standard error of the mean. ^a,b,c^ Means with different superscripts in the same row are different (*p* < 0.05).

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
