# Peer review of "Miúda (Nopalea cochenillifera (L.) Salm-Dyck)—The Best Forage Cactus Genotype for Feeding Lactating Dairy Cows in Semiarid Regions"

_animals, 2021, doi:10.3390/ani11061774_

Round 1

Reviewer 1 Report

The experiment was carried out to determine actual possibility of various cactus forages inclusion to dairy cows’ diet. In order to estimate nutritional value of investigated forages the authors determined chemical composition, apparent digestibility and their effect on cows production performance and milk quality. The applied in the study analytical methods and experiment was designed correctly. However, in my opinion there is a lack of information about ERD of examined cactus forages. It’s well know that feeding the ruminants the same it’s obligatory to provide the same time nutrients available in the rumen for microorganisms. Taking into account that MUN determined in the study vary greatly between treatments. The lower values indicated for not well-balanced when we take into consideration supply of energy and protein available in the rumen what can affect both milk yield and composition. In my opinion the diets should be formulated to be isoenergetic and isoprotein  (including rumen degraded nutrients). Then the answer which cactus forage is better would be more clear. Otherwise if one of the diets components replaced, the conclusion about the value of the forges could be different.

The paper requires linguistic correction, eg. Line 230 – dietary, Line 76 – estimated; 32-33, 48-49 – the same sentence is at the end of the simple summary and summary.

Despite all issues mentioned above I think that the paper meets requirements for this type of works. Therefore, I recommend it for publishing in section of  Animals – Animal Nutrition.

Author Response

Dear reviewers,

Thank you very much for your suggestion and comments. We did our best to accomplish the improvements. The manuscript was once more submitted to a proofreading service. Please, find below the answers for your comments and all modifications are highlighted in the text.

Reviewer 2 Report

This is an interesting manuscript.

I have a few edits/ suggestions.

Be careful extrapolating too much information regarding milk production in a Latin square. I would de-emphazize it a bit.

That said Your MUN values for the GC and MC treatments are low. do you have any in situ data regarding these cacti? As I wonder if the diets were deficient in RDP? Could it be that the synchronization of the N with the NFC was correct? Please address 

Specific edits

61 delete named as

Table 1 subscripts NFC not NCF

137 report as  X g not rpm

173 more information is needed

189 Why FCM instead of ECM, ECM, I believe, considers more variables

234 According to your data MC and SC are the same

325      4.0g/kg?

Author Response

Dear reviewers,

Thank you very much for your suggestion and comments. We did our best to accomplish the improvements. The manuscript was once more submitted to a proofreading service. Please, find attached the answers for your comments and all modifications are highlighted in the text.

Round 2

Reviewer 2 Report

One minor edit- not sure what you mean by "nontriplicated" line 328